# SiGMoiD: A super-statistical generative model for binary data

**Xiaochuan Zhao**[1], **Germán Plata**[2], **Purushottam D. Dixit**[1,3]*

**1** Department of Physics, University of Florida, Gainesville, Florida, United States of America, **2** Elanco Animal Health, Greenfield, Indiana, United States of America, **3** Genetics Institute, University of Florida, Gainesville, Florida, United States of America

* pdixit@ufl.edu

## Abstract

In modern computational biology, there is great interest in building probabilistic models to describe collections of a large number of co-varying binary variables. However, current approaches to build generative models rely on modelers' identification of constraints and are computationally expensive to infer when the number of variables is large ($N{\sim}100$). Here, we address both these issues with **S**uper-stat**i**stical **G**enerative **Mo**del for b**i**nary **D**ata (SiG-MoiD). SiGMoiD is a maximum entropy-based framework where we imagine the data as arising from super-statistical system; individual binary variables in a given sample are coupled to the same 'bath' whose intensive variables vary from sample to sample. Importantly, unlike standard maximum entropy approaches where modeler specifies the constraints, the SiGMoiD algorithm infers them directly from the data. Due to this optimal choice of constraints, SiGMoiD allows us to model collections of a very large number ($N>1000$) of binary variables. Finally, SiGMoiD offers a reduced dimensional description of the data, allowing us to identify clusters of similar data points as well as binary variables. We illustrate the versatility of SiGMoiD using multiple datasets spanning several time- and length-scales.

**Data Availability Statement:** All data and code required to reproduce the analysis in the paper is available on github at https://github.com/zhaoxc099/sigmoid.

**Funding:** XZ and PD would like to thank the University of Florida startup fund for their salaries.

## Author summary

Collectively varying binary variables are ubiquitous in modern biology. Given that the number of possible configurations of these systems typically far exceeds the number of available samples, generative models have become an essential tool in quantitative descriptions of binary data. The state-of-the-art approaches to build generative models have several conceptual limitations. Specifically, they rely on the modeler choosing system-appropriate constraints, which can be challenging in systems with many complex interactions. Moreover, they are computationally expensive to infer when the number of variables is large ($N{\sim}100$). To address this issue, we propose a theoretical generalization of the maximum entropy approach that allows us to model very high dimensional data; at least an order of magnitude higher than what is currently possible. This framework will be a significant advancement in the computational analysis of covarying binary variables.

XZ would like to thank the University of Florida Research Opportunity Fund RPD-ROSF2021 for his salary. The funders had no role in study design, data collection and analysis, decision to publish, or preparation of the manuscript.

**Competing interests:** The authors have declared that no competing interests exist.

## Introduction

Recent technical advances allow us to collect high resolution and high dimensional data across several biological systems. In several cases, these data can be accurately represented as collectively varying binary variables. Significant examples can be found in genomics, where sequencing data is first mapped to gene families and then the presence or absence of thousands of genes across microbial genomes is investigated [1], in microbial ecology, where sequences of the 16s ribosomal gene are first mapped onto operational taxonomic units (OTUs) and then presence or absence of species across microbiomes is investigated to identify direct metabolic interactions [2], or in neuroscience, where electrical current recordings from hundreds of thousands neurons are binarized into spike trains which are then related to organismal level tasks [3].

Unfortunately, estimating the frequency of occurrence of every possible binary configuration from available samples is not possible for any reasonably sized collection; a system with $N$ co-varying binary variables has $2^N$ possible configurations and the number of collected samples is typically orders of magnitude lower than the number of configurations. At the same time, given the complexity of interactions, in most cases, it is infeasible to build bottom-up mechanistic models to describe these systems. A popular alternative is to derive approximate top-down probabilistic models and train those models on the data. Over the past two decades, the maximum entropy (max ent) method [4] has emerged as perhaps the only candidate for building approximate generative models across a variety of contexts [5–12]. Briefly, amongst all probability distributions (models) that are consistent with user-specified constraints, max ent chooses the least biased one; the max ent distribution does not disfavor any outcome unless warranted by the imposed constraints. However, traditional application of max ent has several drawbacks. (1) Perhaps the biggest limitation is that the modeler is required to *a priori* identify constraints that are appropriate for a given system. Depending on the complexity of interactions, these constraints may not be obvious. A work around is to impose a very large number of constraints. For example, a max ent model to analyze correlated firing $N$ neurons will typically involve $N$ constraints on mean firing rates of individual neurons and $\sim N^2$ constraints on covariations in firing rates for all pairs of neurons. (2) These user-identified constraints are imposed using Lagrange multipliers and the multipliers need to be tuned such that model predictions numerically match imposed constraints. However, most often these multipliers cannot be determined analytically and have to be inferred numerically. The most common approach is to use Markov chain Monte Carlo (MCMC) methods to estimate the gradients of the log-likelihood of the data with respect to the Lagrange multipliers and then performing gradient ascent [10,13,14]. These calculations are computationally infeasible even when the number of dimensions is only moderately large (N~100). (3) The numerical values of the imposed constraints are often evaluated using experimental samples which implicitly assumes that samples points are statistically independent of each other. This assumption is not true in most practical applications, for example, in temporally correlated firing of neurons or phylogenetically related protein sequences [10,15]. These limitations taken together have severely limited the application of max ent when there an increasing interest in describing the collective behavior of thousands of cells, genes or microbial species, among others.

In order to study covariation in a large number of binary variables in a constraint-agnostic and numerically efficient manner, we propose a novel dimensionality reduction framework inspired from statistical physics; **S**uper-stat**i**stical **G**enerative **Mo**del for b**i**nary **D**ata (SiGMoiD). SiGMoiD is a generalization of the max ent approach and has several salient features that distinguish it from the state-of-the-art models of binary variables. (1) In SiGMoiD, the modeler only specifies the total number of constraints. The constraints are optimally learned

from the collected samples and a max ent model is fit to those constraints. (2) As a result of this optimal choice of constraints, SiGMoiD requires a much smaller number of constraints than traditional max ent. Consequently, the inference in SiGMoiD is significantly faster than typical max ent models, allowing us to analyze very high dimensional data sets (dimensions $\gg$1000) that remain well out of the reach of current max ent methods. (3) SiGMoiD does not assume that collected samples are drawn from the same distribution. Instead, motivated by superstatistics, it imagines each sample as arising from its own max ent probability distribution. Since each sample is approximated by a small set of Lagrange multipliers, SiGMoiD is also a non-linear dimensionality reduction method. Below, we first sketch the outline of SiG-MoiD and then illustrate it utility by applying it to several data sets.

## Results

### The model

We assume that experimental measurements are in a form where individual samples (data points), indexed by subscript s, comprise $N$ binary variables $\{\sigma_{si}\}$ ($i \in [1,N]$, $s \in [1, S]$) that take values 0 or 1. Let us denote by $\pi_{si}$ the probability that $\sigma_{si} = 1$ and by $\boldsymbol{\pi}_s$ the vector of probabilities $\boldsymbol{\pi}_s = \{\pi_{s1}, \pi_{s2}, \ldots, \pi_{sN}\}$. To motivate our model framework (Fig 1), we imagine the following physical process: for a fixed sample $s$, each binary variable $i$ in the collection of $N$ variables is interacting with the same bath that can exchange $K$ types of extensive variables (energies). The $k^{th}$ type of energy (feature) for each binary variable in the state when it is active ($\sigma_{si} = 1$) is $E_{ki}$ and zero when it is inactive ($\sigma_{si} = 0$) (denoted collectively by $\boldsymbol{E}$). Under these circumstances, the probability of the $i^{th}$ binary variable is equal to 1 in the $s^{th}$ sample is given by the Gibbs-Boltzmann distribution [16]:

$$p(\sigma_{si} = 1) \triangleq \pi_i(\boldsymbol{\beta}_s, \boldsymbol{E}) = \pi_{is} = \frac{\exp(-\sum_k \beta_{sk} E_{ki})}{1 + \exp(-\sum_k \beta_{sk} E_{ki})} \tag{1}$$

In Eq 1, $\boldsymbol{\beta}_s = \{\beta_{s1}, \beta_{s2}, \ldots, \beta_{sK}\}$ are the intensive variables (latent space representation or latents) specific to sample $s$. The probabilities in Eq 1 are the maximum entropy probability distributions when averages $\langle E_{ki} \rangle_s$ ($k \in [1,K]$) of the $K$ types of energies are specified for each variable ($i \in [1,N]$) for every sample $s$.

We have set up the model such that the latents $\beta_{sk}$ depend on the sample index $s$ but not on the index $i$ of the binary variables. In contrast, the features $E_{ki}$ depend on the binary variables $i$ but are shared across all samples $s$. Let us consider that we are given $S$ samples $\{\sigma_{si}\}$ ($i \in [1,N]$, $s \in [1,S]$) of the binary variables. From these samples, we infer sample-specific latents $\boldsymbol{\beta}_s$ and

**Fig 1. Schematic of the SigMoiD approach.** Probabilities $\pi_{si}$ for binary variables $i$ in samples $s$ are generated according to a Gibbs-Boltzmann distribution with energies (features) $\boldsymbol{E}$ and inverse temperatures (latents) $\boldsymbol{\beta}$. The observed data (samples) is assumed to have arisen from Bernoulli trials based on the model probabilities. SiGMoiD infers the parameters $\boldsymbol{E}$ and $\boldsymbol{\beta}$ using maximum likelihood inference.

sample-independent features $E$. To that end, we take a maximum likelihood approach. We write the log-likelihood of the data given the parameters:

$$L = \sum_{i,s} \sigma_{si} \log \pi_{si} + (1 - \sigma_{si})\log(1 - \pi_{si}) \qquad (2)$$

The log-likelihood can be maximized to determine the parameters using gradient ascent. The gradients are given by:

$$\frac{\partial L}{\partial \beta_{sk}} = \sum_{i} (\pi_{si} - \sigma_{si})E_{ki} \text{ and } \frac{\partial L}{\partial E_{ki}} = \sum_{s} (\pi_{si} - \sigma_{si})\beta_{sk} \qquad (3)$$

**SiGMoiD has several salient features.** First, similar to other non-linear dimensionality reduction methods, if $K \ll N$, SiGMoiD offers a reduced dimensional description of the data; the $K$ dimensional vectors $\boldsymbol{\beta}_s$ embed the $N$ dimensional data point $\boldsymbol{\sigma}_s$ in a $K \ll N$ dimensional space. In addition, since SiGMoiD is a fully probabilistic approach, it can also be used as a generative model. Random samples can be generated as follows. We first select a random set of latents $\boldsymbol{\beta}_s$, evaluate the probabilities $\boldsymbol{\pi}_s$ and sample random variables $\boldsymbol{\sigma}$ as Bernoulli variables using those probabilities. SiGMoiD also allows us to evaluate the probability of a new set of binary variables $\boldsymbol{\sigma}$ given the other observations. Specifically, the probability is

$$p(\boldsymbol{\sigma}) = \frac{1}{S} \sum_{s} p(\boldsymbol{\sigma}|\boldsymbol{\beta}_s, \boldsymbol{E}) \qquad (4)$$

where

$$p(\boldsymbol{\sigma}|\boldsymbol{\beta}_s, \boldsymbol{E}) = \prod_{i} \pi_{si}^{\sigma_i}(1 - \pi_{si})^{1-\sigma_i} \qquad (5)$$

is the probability of observing the binary variables $\boldsymbol{\sigma}$ when the latents are fixed at $\boldsymbol{\beta}_s$.

We note that even though we have proposed to identify the parameters using a maximum likelihood approach, given a suitable prior $p_{prior}(\boldsymbol{\beta}, \boldsymbol{E})$ over the parameters, we can also estimate the Bayesian uncertainty in parameter estimation using the posterior distribution

$$p_{post}(\boldsymbol{\beta}, \boldsymbol{E}|\boldsymbol{\sigma}) = p_{prior}(\boldsymbol{\beta}, \boldsymbol{E}) \prod_{i,s} \pi_{si}^{\sigma_{si}}(1 - \pi_{si})^{1-\sigma_{si}}. \qquad (6)$$

Finally, we comment on the degeneracies in SiGMoiD inference procedure. If we multiply the $S \times K$ matrix of latents $\beta_{sk}$ by a $K \times K$ invertible matrix $\boldsymbol{M}:\boldsymbol{\beta} \rightarrow \boldsymbol{\beta M}$ and simultaneously multiply the $K \times N$ matrix of features $E_{ki}$ by $\boldsymbol{M^{-1}}:\boldsymbol{E} \rightarrow \boldsymbol{M^{-1}E}$, the SiGMoiD predictions do not change. Therefore, SiGMoiD- based inference of parameters will significantly depend on the initialization. These degeneracies can therefore be minimized or completely removed without changing model performance by imposing additional restrictions on the parameters, for example, by requiring that the latents or the features are orthogonal to each other, i.e. imposing $\boldsymbol{\beta}^{\mathrm{T}}\boldsymbol{\beta} = \boldsymbol{I_K}$ or $\boldsymbol{EE}^{\mathrm{T}} = \boldsymbol{I_K}$. We leave these explorations for future studies.

## Accuracy of SiGMoiD as a probabilistic model: Modeling the collective firing of neurons

Before illustrating SiGMoiD using high dimensional data sets, we first show a comparison between SiGMoiD and the standard approach to model binary variables; a max ent model. We use a previously collected data set measuring the collective firing of 160 retinal neurons for the duration of a movie that lasted 19 seconds [17,18] (see Supplementary Information). We note

that inference of a max ent model for the collective firing of all 160 neurons is currently computationally prohibitive. We chose the 15 most active neurons in the data (15 highest firing propensities) to illustrate our approach. First, we inferred a max ent model from the data that constrained mean firing rates and pairwise correlations. The max ent model describes the probability of any configuration $\boldsymbol{\sigma}$ as:

$$p(\boldsymbol{\sigma}) \propto \exp\left(-\sum_{i,j} J_{ij}\sigma_i\sigma_j\right) \tag{7}$$

In Eq 7, $J_{ij}$ are coupling constants (Lagrange multipliers) that need to be inferred from the data, typically using gradient ascent of the log likelihood of the data [10,13,14]. Given that there are only $2^{15} \sim 3 \times 10^4$ states for 15 neurons, we could estimate model predictions and therefore the coupling constants by an exhaustive brute force summation over all possible states without resorting to MCMC simulations to estimate the gradients. This minimized the errors in max ent inference that arise due to inaccuracies in MCMC-based estimates of average firing rates and neuron-neuron correlations. The resulting max ent model perfectly captured the average firing rates and the pair correlations (S1 Fig).

In Fig 2, we compare the max ent model with SiGMoiD. The max ent model has $\binom{15}{2} =$ 105 neuron-specific parameters. To match that number, we choose $K = 7$ in SiGMoiD. In panel (a), we show a comparison between the raw frequencies of individual configurations obtained from data (x-axis) to model predicted probabilities (y-axis, red: max ent, blue: SiG-MoiD). The raw frequencies were obtained by counting the number of instances of individual configurations across all $10^4$ samples. It is clear that SiGMoiD has smaller error compared to the max ent model (mean absolute error $8.6 \times 10^{-6}$ vs $1.4 \times 10^{-5}$). In panel (b), we plot the probability that $n$ neurons fire at the same time as observed in the data (black), predicted using SiG-MoiD (blue), and using the max ent model (red). Here too, the SiGMoiD model performs well when capturing the probability of simultaneous activity. In panels (c) and (d), we plot the absolute values of the three-body correlations $|\langle\delta\sigma_i\delta\sigma_j\delta\sigma_k\rangle|$ as observed in the data (x-axis) and as predicted by the model (y-axis, SiGMoiD, panel (c), max ent, panel (d)). Both models capture the three body correlations with reasonable accuracy; the mean absolute error is $8.2 \times 10^{-4}$ vs $1.4 \times 10^{-3}$ for the SiGMoiD and the max ent model respectively. This analysis illustrates that the SiGMoiD is better than the max ent based model at capturing the data and making predictions. Next, we move to systems that are currently well out of the reach of max ent methods.

## Inference of interactions from bacterial co-occurrences using SiGMoiD

Gut microbiomes are complex ecosystems whose statistical properties have received significant attention in the last couple of years [19,20]. Gut bacteria live in species-rich communities where they compete for nutrients and also exchange metabolites with each other. Describing these interactions is critical to map the ecological networks of gut microbiomes and identify targets for controlling microbial communities [21]. However, many of the direct metabolic interactions between gut microbes are likely to occur at a micron length scale [2]. Therefore, it is infeasible to infer these interactions from macroscopic, community-wide abundance co-variation.

To address this issue, Sheth et al. [2] recently probed the spatial organization of the gut microbiome at the micron length scale, allowing them to capture putative direct interactions between bacteria. In these experiments, Sheth et al. [2] fractionated mice guts into particles with a median diameter of 30 $\mu m$ and quantified the membership of 347 operational

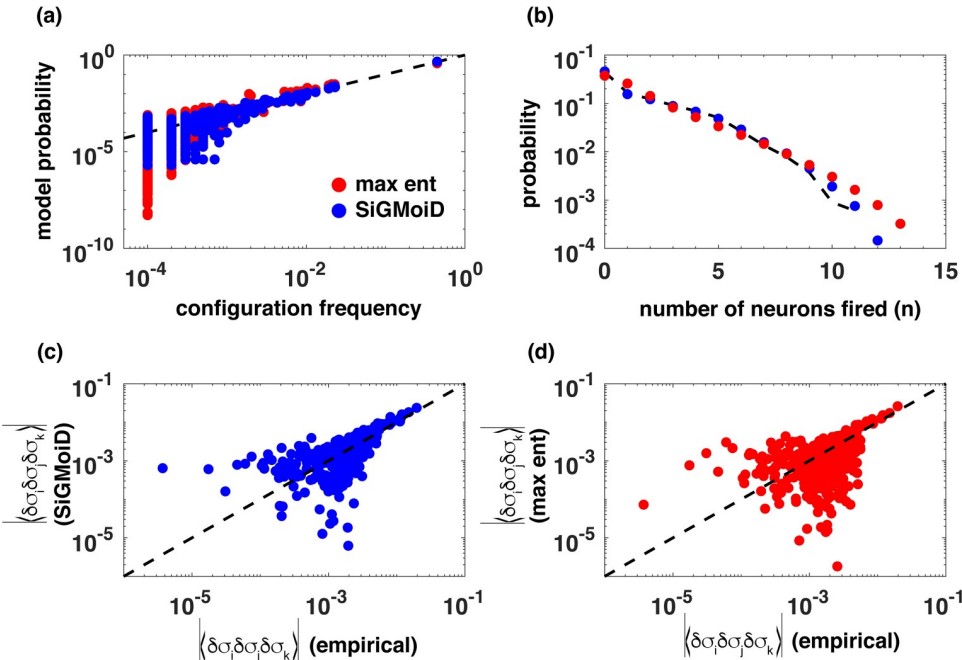

**Fig 2. Comparison of SiGMoiD with max ent modeling. (A)** the frequencies of individual configurations estimated from the samples (x-axis) and from the two models (y-axis), (red: max ent, blue: SiGMoiD). Only the frequencies of the 1442 configurations observed at least once in the samples are shown. **(B)** the probability that $n$ neurons fire in any given configuration as estimated from samples (black), the max ent model (red), and SiGMoiD (blue), **(C)** and **(D)** comparison between the absolute values of three variable correlations $\langle \delta\sigma_i \delta\sigma_j \delta\sigma_k \rangle$ estimated from data (x-axis) and those using the models (y-axis). There are $\binom{15}{3} = 455$ such correlations.

taxonomic units (OTUs) across 1406 particles. However, given that co-occurrences are transitive (if A interacts and co-occurs with B, and B interacts and co-occurs with C, then A co-occurs with C even in the absence of interactions), it is not possible to use simple co-occurrence calculations to identify putative pairs of directly interacting OTUs [22].

Given that SiGMoiD can directly model occurrence of individual OTUs across particles, it can be used to identify clusters of OTUs that co-vary across particles as well as clusters of particles that show specific OTU occurrence profiles. We therefore analyzed the data collected by Sheth et al. [2] using SiGMoiD (see Supplementary Information). Each particle was characterized by a binary vector of dimension representing the OTUs present in that particle. It is evident that SiGMoiD will fit the data better as the number of components $K$ increases. Unlike the neuron samples which were correlated in time and across different trials, the microbiome particles are likely to be closer to statistical independence. Therefore, we can use information theory-based criteria to select the optimal $K$ that fits the data but avoids overfitting. In S2 Fig, we show the Akaike information criteria (AIC) vs. $K$ for the OTU data. The model picks out $K = 8$ as the optimal value which we use in further analysis. When individual samples are correlated, one can use cross-validation; splitting the samples in a training vs. a validation set and then evaluating the probability of the validation set, to avoid overfitting.

The number of species found in each particle, a gross descriptor of the complexity of the community [8], varies substantially from particle to particle. As shown in Fig 3, generative modeling of the particles using SiGMoiD accurately captures this quantifier of ecological

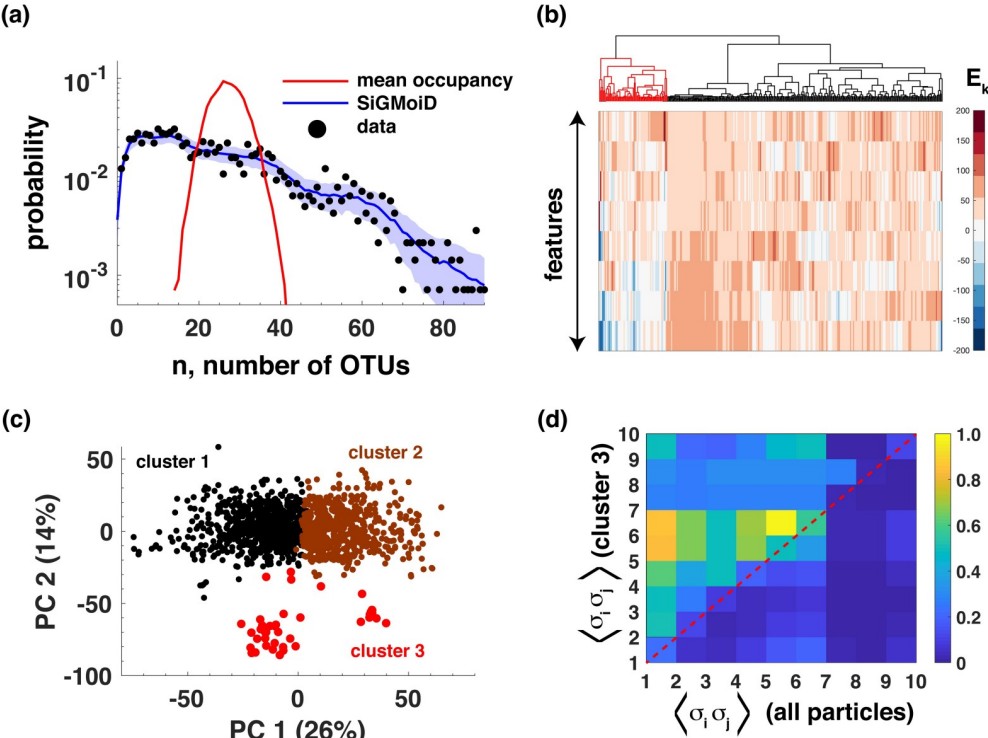

**Fig 3. SiGMoiD models bacterial co-occurrences and interactions.** (**A**) the probabilities of co-occurrence of multiple OTUs in a single particle. Black circles represent the data, the blue line and shaded blue region represents the SiGMoiD predictions and standard deviations around the predictions, and the red line represents a prediction based on mean occupancies of OTUs. (**B**) Clustergram showing similarity in features between OTUs. The identified outgroup is marked red. (**C**) PCA of 3 clusters identified using particle-specific latents $\boldsymbol{\beta}_s$. (d) (upper half) Co-occurrence frequencies of 10 OTUs whose occurrence frequency was most significantly different in cluster 3 compared to the baseline co-occurrence frequencies of the same OTUs (bottom half).

complexity. In Fig 3A we show the probability of co-occurrence of multiple OTUs in any community as observed in the data (black circles) and as predicted by SiGMoiD (blue line). We compare these distributions with the null expectation given by the probabilities of occurrence of $n$ OTUs in any given particle calculated using the occurrence frequencies of individual OTUs but neglecting the correlations between OTUs (red line). The significant difference between the two suggests that SiGMoiD can accurately capture the interactions between OTUs, which in turn allows it to predict the co-occurrence distribution.

In fact, SiGMoiD can be used to identify specific bacteria with similar occurrence profiles across particles. SiGMoiD characterizes each binary variable (here, OTU presence/absence) using a $K$ dimensional vector of features. OTUs with similar features will have similar co-occurrence profiles as well. Therefore, the feature vector can be used to identify clusters of co-occurring OTUs. SiGMoiD-based clustering of OTUs is a more direct way of identifying clusters by relying on inferred inherent properties of the OTUs rather than their co-occurrence profiles. Fig 3B shows a hierarchical clustering plot of all OTUs using SiGMoiD-inferred features. Among the several identified clusters, we focus on the cluster of 69 OTUs highlighted in the figure. The gut microbiome of mice is dominated by OTUs belonging to the family *Lachnospiraceae*; ~53% of all the OTUs in the analyzed data belonged to this family. However, these OTUs are not equally distributed across the particles. The cluster highlighted in the figure is statistically significantly enriched with the family *Lachnospiraceae* (46 out of 69, single

tailed hypergeometric distribution p-value 0.009). Notably, the OTUs belonging to this cluster had predominantly positive correlations across different particles; 2330 out of the 2346 unique pairs had a positive correlation with 92% of pairs with a p-value less than $10^{-2}$ (84% of pairs with a p-value less than $10^{-4}$ and 74% of pairs with a p-value less than $10^{-6}$). In comparison, only 50% of unique pairs from other OTUs had a positive correlation and only 23% of those correlations had a p-value less than 0.01. These analyses suggest that SiGMoiD-based features can identify clusters of OTUs that significantly co-occur in a given ecology. There are two types of metabolic interactions between bacteria that lead to co-occurrence in an ecosystem [23], especially at the micron length scale [2]. Genetically related bacteria tend to co-occur because they have similar metabolic networks and can compete for the same resources. In contrast, genetically dissimilar bacteria have different metabolic networks and can cross-feed each other; one species utilizing the metabolic byproducts of another. Therefore, this cluster likely represents the co-occurrence of multiple species in the *Lachnospiraceae* family that compete with each other for the same resources in the mouse gut.

In addition to identifying OTUs that have similar occurrence profiles across communities, SiGMoiD can also be used to identify communities that have similar OTU occurrence profiles. SiGMoiD embeds each high dimensional binary sample in a much lower dimensional space of sample-specific $\boldsymbol{\beta}$ latents. Using K-means clustering of sample-specific $\boldsymbol{\beta}$ latents, we identified 3 clusters of particles (S2 Fig). Principal component analysis (PCA)-based visualization of the particles clearly shows the three identified clusters (Fig 3C). Notably, several specific OTUs were co-present with much higher occurrence frequencies in the identified small cluster (cluster 3, comprising 47 particles). In Fig 3D, we compare the pairwise co-occurrence frequency of 10 OTUs whose occurrence frequency was identified to be most significantly different between particles in cluster 3 compared to the baseline using a hypergeometric test (S1 Table). It is clear that compared to the baseline co-occurrence frequency (sub-diagonal half of Fig 3D), the pairwise co-occurrence frequencies of the 10 OTUs are significantly elevated in the communities in cluster 3. These analyses show that SiGMoiD can also identify specific communities that comprise strongly co-occurring bacteria that differentiate them from other communities. These significant clusters can potentially be investigated for direct co-operative or competitive interactions, as well as their association with distinct regions of the gut. Importantly, clusters of particles with these tightly correlated species were not detected when we clustered the samples (binary vectors) directly using the same approach (S3 Fig).

## Identifying missing metabolic reactions using SiGMoiD

The metabolic repertoire of microorganisms enables them to convert nutrients into biomass and energy and underlies phenotypic traits central to their ecosystem roles [24]. E.g. microbial fermentation in the gut and its impact on human health [25] or methane production in animal agriculture or wetland ecosystems [26,27].

Genome sequencing and annotation methods have enabled the identification of metabolic transformations that individual microbes can potentially carry out through the reconstruction of their metabolic networks. Metagenomics sequencing on the other hand, has allowed the study of the genomes and metabolic properties of microorganisms in microbiomes of interest which have not yet been cultured and characterized. Nevertheless, due to the complexity of these microbial communities, it is often not possible to determine the full genomic content of most members of a given microbiome. Therefore, beyond a few highly abundant microbes whose genomes can be inferred to a reasonable degree of completeness from metagenomics data, the metabolic capabilities of many microbes of interest can only be partially assessed through metagenome annotation and binning methods. Here we show that SiGMoiD can be

used to infer missing reactions in the metabolic repertoire of incompletely sequenced genomes, as are often produced by metagenome assembly and binning pipelines.

To that end, we downloaded genome-scale metabolic reconstructions for ~4000 bacteria from KBase [28] generated with the ModelSeed pipeline [29] (see Supplementary Information). Often, intercompartmental transport reactions and other reactions are added to metabolic reconstructions to ensure mass balance and viability of biomass production without a clear identification of the genes that may carry out these reactions. To avoid biasing our approach towards or against these *ad hoc* additions, we only retained those reactions that were assigned to one or more genes in the reconstructions. This resulted in a total of ~3300 reactions across all bacterial metabolic reaction sets. We randomly selected 400 bacteria as a test set to quantify the accuracy of our predictions. We inferred SiGMoiD parameters on the rest of the bacteria and used the inferred parameters to identify missing reactions in the test dataset as follows.

First, for each bacteria in the testing set, we removed a fixed fraction of reactions ranging from 10% to 90% to simulate incomplete genome coverage from metagenomic sequencing. Next, we used SiGMoiD and the known reactions for each bacteria in the testing set to predict the missing reactions. We used $K = 15$ components. We identified groups of reactions with similar occurrence profiles across the bacterial world by clustering their corresponding $E_{ki}$ features obtained from the training data using agglomerative hierarchical clustering [30]. The similarity $d_{ij}$ between two metabolic reactions $i$ and $j$ was defined as the L2 norm: $d_{ij}^2 = \sum_k (E_{ki} - E_{kj})^2$. Hierarchical clustering divides the reactions into multiple clusters depending on a tunable level of clustering. On one end, all reactions are clubbed into a single cluster, and on the other end, every reaction is its own cluster. For any given level of clustering, we employed a simple rule to predict missing reactions from known reactions. If any one of the reactions in a cluster is known to be present in the metabolic repertoire of a bacterium, all other reactions in the cluster are also predicted to be in the network. Using this simple prediction model and by varying the level of clustering, we obtained true and false positive rates for the predictions for each bacteria. These rates were averaged across all bacteria for a given level of clustering and plotted as a receiver operating characteristic (ROC) curve. Fig 4A shows that this simple approach to predict the missing metabolic reactions performs exceedingly well. The area under the curve when only 10% of the reactions are known is 0.916, which increases to 0.988 when 90% of the reactions are known. Notably, the performance of our approach is extremely

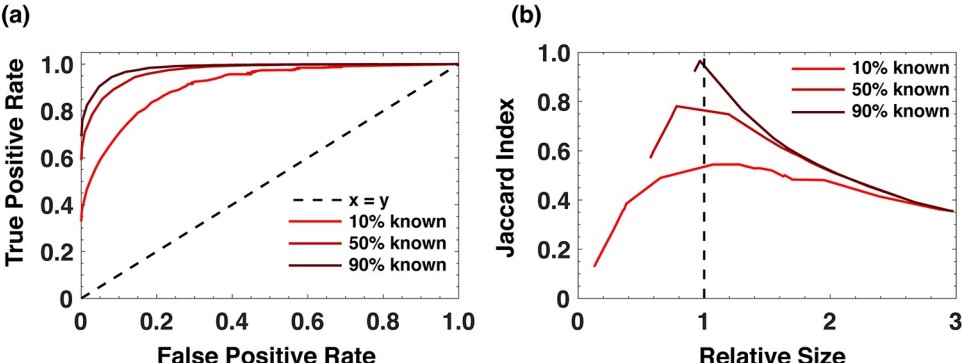

**Fig 4. SiGMoiD predicts the presence/absence of metabolic reactions.** (**A**) Receiver operating characteristic (ROC) curve for SiGMoiD-based prediction of missing metabolic reactions. Different lines represent metabolic models with different fractions of known reactions. (**B**) The mean Jaccard index between the set of predicted metabolic reactions and the actual metabolic reactions in any species (y-axis) vs. the relative size of the predicted network to the actual network (x-axis).

accurate across different values of $K$ and different fractions of missing reactions (S4 Fig). Importantly, the highest overlap between the true metabolic repertoire and the predicted metabolic repertoire as quantified by the Jaccard index occurs near the true size of the network (Fig 4B).

Notably, unlike other gene inference methods that rely on phylogenetic placement of the query genome against a reference database [31,32] or approaches that can use phenotypic knowledge about the organism (nutrients that can sustain its growth, biomass composition, etc.) [33] our approach predicts reactions solely based on their co-occurrence profile across the bacterial kingdom. Therefore, our approach can be integrated with other methods to robustly predict missing reactions in genome scale metabolic reconstructions.

## Discussion

A deluge of biophysical data in the last decade has called for the development of top-down modeling approaches. Here, instead of describing the data from first principles mechanistic models, one constructs probability distributions that represent it. As a result, generative models of collective behavior have become essential to modeling several biophysical systems. The most popular way to generate top-down models is the maximum entropy (max ent) approach wherein one approximates the data using a probabilistic model that reproduces lower order statistics estimated from the data. The max ent approach has the significant conceptual advantage that it represents the simplest model consistent with the imposed constraints. However, there are two significant drawbacks. First, the constraints are hand-picked by the modeler and the model therefore depends on these constraints. For binary data, constraints of averages and pair correlations have become popular. Second, the inference of max ent models for large data sets can be computationally expensive and it may be unrealistic to infer models for >100 binary variables.

To address these issues, we developed SiGMoiD. SiGMoiD takes an agnostic approach about the constraints. In SiGMoiD, instead of specifying the constraints, the user only specifies the total number of constraints. SiGMoiD learns these constraints from the data. Moreover, parameter inference in SiGMoiD is orders of magnitude faster than max ent inference. We showed using three data sets of varying complexity that SiGMoiD not only performs as well as max ent models in terms of accuracy but can also be applied to study very large data sets that are currently out of the reach of max ent inference. Going forward, we believe that this computationally efficient and conceptually straightforward approach will be immensely valuable in modeling collective behavior of high dimensional data.

We have previously developed SiGMoiD-like approaches [16,34] to model multinomially distributed abundance data common in sequencing studies including 16s sequencing based characterization of the microbiome [34]. Going forward, the most straightforward generalization to SiGMoiD is applying it to study amino acid/nucleotide variation in sequencing data. Concretely, a collection of $N$ protein sequences of length $L$ each can be represented by binary variables $\sigma_{nia} = 0/1$ where $n \in [1, N]$ is the index of the sample, $i \in [1, L]$ is the position of the amino acid in the protein sequence, and $a \in [1, 21]$ is the identity of the amino acid (there are 20 naturally occurring amino acids, plus an additional index for a gap in the multiple sequence alignment). The probability $\pi_{nia}$ of observing amino acid $a$ in position $i$ in the $n^{\text{th}}$ sequence can be modeled using a tensor-based decomposition as was recently done in the analysis of variability in the microbiome [35]:

$$\pi_{nia} \propto \exp\left( - \sum_{k_1 k_2 k_3} \beta_{nk_1} E_{nk_2} A_{nk_3} Z_{k_1 k_2 k_3} \right). \tag{8}$$

We leave developing this generalization to future work.

## Supporting information

**S1 Fig. Mean firing rates and correlations as observed in the data (x-axis) and as predicted by the max ent model (y-axis).**
(TIF)

**S2 Fig.** (A) Akaike information criterion as a function of K, the number of components used to model the microbiome co-occurrence data. (B) Mean silhouette score as a number of clusters using K-means clustering of the particle-specific βs latents.
(TIF)

**S3 Fig. n = 3 clusters of particles were identified using K-means clustering of the microbiome co-occurrence, shown here using the first two principal components of the particle-specific βs latents.**
(TIF)

**S4 Fig. Performance of the SiGMoiD-based approach to identify missing metabolic reactions with K = 20 components (panels a and b) and K = 40 components (panels c and d).**
(TIF)

**S1 Table. Genera of OTUs that are most enriched in cluster 3 using a hypergeometric test and the corresponding p-values.**
(DOCX)

**S1 File. Information about curating the data used in this analysis.**
(DOCX)

## Author Contributions

**Conceptualization:** Xiaochuan Zhao, Purushottam D. Dixit.

**Data curation:** Xiaochuan Zhao, Germán Plata, Purushottam D. Dixit.

**Formal analysis:** Xiaochuan Zhao, Purushottam D. Dixit.

**Funding acquisition:** Purushottam D. Dixit.

**Investigation:** Xiaochuan Zhao, Germán Plata, Purushottam D. Dixit.

**Methodology:** Xiaochuan Zhao, Germán Plata, Purushottam D. Dixit.

**Project administration:** Purushottam D. Dixit.

**Resources:** Xiaochuan Zhao, Germán Plata, Purushottam D. Dixit.

**Software:** Xiaochuan Zhao, Purushottam D. Dixit.

**Supervision:** Germán Plata, Purushottam D. Dixit.

**Validation:** Xiaochuan Zhao, Purushottam D. Dixit.

**Visualization:** Xiaochuan Zhao, Purushottam D. Dixit.

**Writing – original draft:** Purushottam D. Dixit.

**Writing – review & editing:** Xiaochuan Zhao, Germán Plata, Purushottam D. Dixit.

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
