## [Decision Letter · Decision Letter 0]

24 May 2021

Dear Dr. Dixit,

Thank you very much for submitting your manuscript "SiGMoiD: A super-statistical generative model for binary data" for consideration at PLOS Computational Biology.

As with all papers reviewed by the journal, your manuscript was reviewed by members of the editorial board and by several independent reviewers. In light of the reviews (below this email), we would like to invite the resubmission of a significantly-revised version that takes into account the reviewers' comments.

As you will see from the reports, both referees are generally favorable and find the work of potential interest, but they raise important issues that we must ask you to address, in the form of a revised manuscript, before we reach a final decision on publication. In particular, it seems essential to address Referee 1's concerns about relation to prior work and the applicability of the method to different types of data. Referee 2 has also raised several issues about the method design that must be addressed. Please ensure that these and all other issues raised by the referees are addressed in full if you decide to submit a revised version of the manuscript.

We cannot make any decision about publication until we have seen the revised manuscript and your response to the reviewers' comments. Your revised manuscript is also likely to be sent to reviewers for further evaluation.

Sincerely,

Joshua Welch

Guest Editor

PLOS Computational Biology

Jian Ma

Deputy Editor

PLOS Computational Biology

As you will see from the reports, both referees are generally favorable and find the work of potential interest, but they raise important issues that we must ask you to address, in the form of a revised manuscript, before we reach a final decision on publication. In particular, it seems essential to address Referee 1's concerns about relation to prior work and the applicability of the method to different types of data. Referee 2 has also raised several issues about the method design that must be addressed. Please ensure that these and all other issues raised by the referees are addressed in full if you decide to submit a revised version of the manuscript.

Reviewer's Responses to Questions

**Comments to the Authors:**

Reviewer #1: My review is attached in the form of a separate document.

Reviewer #2: Zhao, Plata and Dixit present a novel method to characterize binary data. They apply their method to three experimental datasets and demonstrate its ability to capture nontrivial trends in the context of neuroscience and bacterial populations. Overall, the results seem promising and distinct from the go-to, top-down approach of maximum entropy to model experimental data. I have a few questions about the details of SiGMoiD and how it distinguishes itself from maximum entropy to hopefully clarify SiGMoiD’s novelty in modeling binary data.

1. The authors mention several times that SiGMoiD parameter estimation is much faster than that of maximum entropy. Implicit in this discussion seems to be that maxent methods require Markov chain Monte Carlo (MCMC) simulations to estimate parameters while SiGMoiD can employ gradient ascent. First, why is gradient ascent necessarily faster than MCMC in this case? Second, why can maxent not be cast in the same framework to have parameters estimated by gradient ascent?

Could the authors elaborate why maxent using MCMC cannot infer models greater than 100 variables? Are the authors assuming that the maxent model corresponds to one that enforces constraints on the covariances as well as the means?

Given that both energies and betas are fitted in SiGMoiD, I would think that there would be a lot of degeneracies in the likelihood, potentially making parameter estimation challenging. Could the authors comment on this potential issue?

2. Could the authors comment why they limit SiGMoiD to only mean constraints? Why not also look at pairwise constraints, for example? The fact that SiGMoiD currently only encodes mean constraints needs to be delineated in the main text because statements regarding SiGMoiDs ability to infer any constraints from the data are more limited that what a quick read might imply.

3. I am curious why SiGMoiD is limited to binary data. What steps in the SiGMoiD pipeline are not possible to numerically compute if samples are not binary?

More comments:

Have the authors considered comparing SiGMoiD to a standard clustering algorithm? Just as the authors elegantly compared SiGMoiD with maxent based on pairwise correlations, it would be interesting to see how it performs with standard clustering algorithms, given that clustering is a powerful insight provided by SiGMoiD.

Perhaps to make it clear how the maxent model works and better elucidate its difference between SiGMoiD, the authors should demonstrate that pairwise correlations are perfectly captured by the maxent model in Figure 2.

Discussions in the Introduction on modeling neuron firing should include additional citations to William Bialek’s seminal works, such as (Schneidman et al. 2006; Tkačik et al. 2015)

Perhaps the authors would be interested in the following paper, which proposes a method to infer maxent parameters in large systems (Weistuch et al. 2020).

In the Results, the following sentence which starts as ‘To that end, for any given K …’ has a typo as it abruptly ends with the word ‘of’.

References

Schneidman E, Berry MJ, Segev R, Bialek W. 2006. Weak pairwise correlations imply strongly correlated network states in a neural population. Nature 440:1007–1012.

Tkačik G, Mora T, Marre O, Amodei D, Palmer SE, Berry MJ, Bialek W. 2015. Thermodynamics and signatures of criticality in a network of neurons. Proc. Natl. Acad. Sci. U. S. A. 112:11508–11513.

Weistuch C, Agozzino L, Mujica-Parodi LR, Dill K. 2020. Inferring a network from dynamical signals at its nodes. PLoS Comput. Biol. 1:1–18.

**Have the authors made all data and (if applicable) computational code underlying the findings in their manuscript fully available?**

Reviewer #1: Yes

Reviewer #2: Yes

PLOS authors have the option to publish the peer review history of their article (what does this mean?). If published, this will include your full peer review and any attached files.

Reviewer #1: No

Reviewer #2: No
---

## [Decision Letter · Decision Letter 1]

30 Jun 2021

Dear Dr. Dixit,

Thank you very much for submitting your manuscript "SiGMoiD: A super-statistical generative model for binary data" for consideration at PLOS Computational Biology. As with all papers reviewed by the journal, your manuscript was reviewed by members of the editorial board and by several independent reviewers. The reviewers appreciated the attention to an important topic. Based on the reviews, we are likely to accept this manuscript for publication, providing that you modify the manuscript according to the review recommendations.

Sincerely,

Joshua Welch

Guest Editor

PLOS Computational Biology

Jian Ma

Deputy Editor

PLOS Computational Biology

[LINK]

Reviewer's Responses to Questions

**Comments to the Authors:**

Reviewer #1: The review is uploaded as an attachment.

Reviewer #2: I appreciate the authors’ thorough responses to my questions and clearing up my confusions. I have no further reservations about the current version of the paper. Nice work!

**Have the authors made all data and (if applicable) computational code underlying the findings in their manuscript fully available?**

Reviewer #1: Yes

Reviewer #2: Yes

PLOS authors have the option to publish the peer review history of their article (what does this mean?). If published, this will include your full peer review and any attached files.

Reviewer #1: No

Reviewer #2: No

Figure Files:

Data Requirements:

Reproducibility:

References:

---

## [Editor Report · Decision Letter 2]

13 Jul 2021

Dear Dr. Dixit,

We are pleased to inform you that your manuscript 'SiGMoiD: A super-statistical generative model for binary data' has been provisionally accepted for publication in PLOS Computational Biology.

Best regards,

Joshua Welch

Guest Editor

PLOS Computational Biology

Jian Ma

Deputy Editor

PLOS Computational Biology

---

## [Editor Report · Acceptance letter]

3 Aug 2021

PCOMPBIOL-D-21-00642R2 

SiGMoiD: A super-statistical generative model for binary data

Dear Dr Dixit,

I am pleased to inform you that your manuscript has been formally accepted for publication in PLOS Computational Biology. Your manuscript is now with our production department and you will be notified of the publication date in due course.

With kind regards,

Katalin Szabo
